# Modelling the Influence of Climate and Vector Control Interventions on Arbovirus Transmission

**DOI:** 10.3390/v16081221

**Published:** 2024-07-30

**Authors:** Emma L. Fairbanks, Janet M. Daly, Michael J. Tildesley

**Affiliations:** 1The Zeeman Institute for Systems Biology & Infectious Disease Epidemiology Research, Mathematics Institute and School of Life Sciences, University of Warwick, Coventry CV4 7AL, UK; 2One Virology—Wolfson Centre for Global Virus Research, School of Veterinary Medicine and Science, University of Nottingham, Loughborough LE12 5RD, UK

**Keywords:** modelling, vectorial capacity, African horse sickness, bluetongue, Schmallenberg, epizootic haemorrhagic disease, basic reproduction number, *Culicoides*

## Abstract

Most mathematical models that assess the vectorial capacity of disease-transmitting insects typically focus on the influence of climatic factors to predict variations across different times and locations, or examine the impact of vector control interventions to forecast their potential effectiveness. We combine features of existing models to develop a novel model for vectorial capacity that considers both climate and vector control. This model considers how vector control tools affect vectors at each stage of their feeding cycle, and incorporates host availability and preference. Applying this model to arboviruses of veterinary importance in Europe, we show that African horse sickness virus (AHSV) has a higher peak predicted vectorial capacity than bluetongue virus (BTV), Schmallenberg virus (SBV), and epizootic haemorrhagic disease virus (EHDV). However, AHSV has a shorter average infectious period due to high mortality; therefore, the overall basic reproduction number of AHSV is similar to BTV. A comparable relationship exists between SBV and EHDV, with both viruses showing similar basic reproduction numbers. Focusing on AHSV transmission in the UK, insecticide-treated stable netting is shown to significantly reduce vectorial capacity of *Culicoides*, even at low coverage levels. However, untreated stable netting is likely to have limited impact. Overall, this model can be used to consider both climate and vector control interventions either currently utilised or for potential use in an outbreak, and could help guide policy makers seeking to mitigate the impact of climate change on disease control.

## 1. Introduction

Vectorial capacity, defined as the total number of potentially infectious bites that would arise from all the vectors biting a single infectious host on a single day [1], is a concept used in epidemiology to measure the potential ability of a vector species to transmit a pathogen to a susceptible host population. There are many models for vectorial capacity, each considering different bionomic parameters [2,3,4,5].

The rate at which vectors complete the gonotrophic cycle, defined as the time required for locating a host, blood feeding, egg maturation and oviposition, has been found to be temperature-dependent [6,7,8,9,10]. The extrinsic incubation period (EIP), defined as the time required for a vector to become infectious after consumption of an infected blood meal, is also temperature-dependent [11,12]. Multiple models consider how vectorial capacity changes with different climatic conditions [13,14,15,16,17,18,19,20,21]. These are usually used to assess variations in risk geographically and/or for climate change scenarios. Brand and Keeling [13] suggested a model for vectorial capacity of bluetongue virus (BTV), a *Culicoides*-borne orbivirus of veterinary significance, which considered the temperature dependence of the gonotrophic cycle length, vector mortality, and the EIP.

Chitnis et al. [22] suggested a vectorial capacity model for malaria transmission derived from a discrete-time entomological model of the *Anopheles* feeding cycle. Updated versions of this model have been utilised to predict the community-level effect of using a new vector tool on *Plasmodium* transmission [23,24,25]. The model considers the effects of vector-control interventions at different stages during the mosquito feeding cycle. Although most of the parameter values of the model are measurable, several, such as the ovary sac proportion (proportion of mosquitoes with uncontracted ovary sacs) require technical time-consuming experiments; therefore, there are limited data available on these parameters. One advantage of this model is that it incorporates host selection, which is influenced by the availability of competent and non-competent hosts and the vector’s preferences for these hosts. This consideration is quantified using the human blood index, defined as the proportion of vectors that have fed on humans relative to the total number of blood-fed vectors analysed.

We suggest an updated version of the Brand and Keeling [13] model that considers the influence of climate on vectors, as well as features from the Chitnis et al. [22] model, namely host selection and the impact of vector control tools at various stages of the gonotrophic cycle. This model considers both personal protection through reduction in feeding, as well as community protection through vector mortality and disarming (feeding inhibition due to intoxication). This model is then parameterised for *Culicoides*-borne arboviruses of veterinary importance in Europe: BTV, African horse sickness virus (AHSV), Schmallenberg virus (SBV), and epizootic haemorrhagic disease virus (EHDV). This model is then applied to temperature data from the United Kingdom, which has previously seen BTV and SBV outbreaks. The potential reduction in vectorial capacity for AHSV is then calculated for insecticide-treated stable netting, dependent on behaviour of owners.

Feeding multiple times per gonotrophic cycle is observed to occur naturally for some vectors (for example, *Aedes* mosquitoes), increasing opportunities for disease transmission [26]. This behaviour has not been investigated for *Culicoides*. In other cases, when vectors are disturbed during a blood meal, they may seek more blood from the same or another host [1]. This model considers the influence of the average number of blood meals per gonotrophic cycle on the predicted vectorial capacity and the impact of vector-control tools.

## 2. Methods

### 2.1. Model Development

Parameters for the model are given in Table 1.

This model, similarly to the Brand and Keeling [13] model, allows for several variables to be dependent on climate. Climatic-dependent variables will be considered as vectors, with each entry representing a predicted daily value, calculated from climatic data. Included in these variables are the vector gonotrophic cycle completion rate, vector mortality rate, and the pathogen incubation rate. Alternatively, following the approach of the Chitnis et al. [22] model, these variables can be treated as constants in regions with relatively homogeneous climates or when data are insufficient. Under these conditions, all elements of the vectors would be identical.

Given that a pathogen is transmitted to a vector on day *t*, the probabilities that the vector survives until and is infectious on day τ are
(1)$$P_{\mu }\left(t,\tau \right)=\exp\left(-\sum_{i=t}^{\tau }\hat{\mu }\left(i\right)\right)\;\mathrm{and}$$
(2)$$P_{\sigma }\left(t,\tau \right)=\Gamma_{k}\left(\sum_{j=t+1}^{\tau }\hat{\sigma }\left(j\right)\right)$$
respectively, where $\Gamma_{k}$ is the cumulative distribution function of a gamma distribution with mean 1 and shape k. For the survival probability (Equation (1)), the accumulated rate includes the infection day, because the vector must survive that day to transmit the pathogen to other hosts. For the probability of the vector completing the EIP (Equation (3)), the accumulated rate starts on the day after infection. This is because most bites occur at sunset and twilight [27]; therefore, we assume the EIP only depends on climatic conditions during days after the bite has occurred. It is important to note that bites occurring in the early hours of the morning are typically counted as part of the previous day’s activity. The temperature during the day affects whether *Culicoides* will be active and bite that night. In this study, we assume that k=1; therefore, we have
(3)$$P_{\sigma }\left(t,\tau \right)=1-\exp\left(-\sum_{j=t+1}^{\tau }\hat{\sigma }\left(j\right)\right),$$
which is a Markovian model.

In the model, when vectors encounter a host, they either bite, are killed preprandially (before feeding), are disarmed, or seek another host. If they bite, they may be killed postprandially (after feeding). Here, we refer to biting, preprandial mortality, and disarming as host-encountering events. The probability of a host-countering event on day *s* is considered to be Markovian, calculated as
(4)$$P_{\alpha }\left(s\right)=1-\exp\left(-\left(\epsilon \hat{\alpha }\left(s\right)\left(\chi \left(\psi \left(1-\phi \left(\pi -\kappa_{M}-\kappa_{D}\right)\right)+\left(1-\psi \right)\right)+\left(1-\chi \right)\right)\right)\right),$$
where parameters are defined in Table 1. Here, the impact of vector-control tools is scaled by their usage (ϕ), coverage (ψ) and the proportion of the vectors that would bite the host type in the absence of the vector-control tool. In this study, we assume that vector control tools are applied only to competent hosts. Vectors that only feed once during their gonotrophic cycle stop host seeking after host-encountering events.

The probability a bite with transmission of a pathogen from a host to a vector and vector to host on days *t* and τ are
(5)$$P_{h\to v}\left(t\right)=P_{\alpha }\left(t\right)\rho_{h\to v}\frac{\chi \left(\psi (1-\phi \pi )+(1-\psi )\right)}{\chi \left(\psi \left(1-\phi \left(\pi -\kappa_{M}-\kappa_{D}\right)\right)+\left(1-\psi \right)\right)+\left(1-\chi \right)}$$
and
(6)$$P_{v\to h}\left(t,\tau \right)=P_{\alpha }\left(\tau \right)P_{\sigma }\left(t,\tau \right)\rho_{v\to h}\frac{\chi \left(\psi (1-\phi \pi )+(1-\psi )\right)}{\chi \left(\psi \left(1-\phi \left(\pi -\kappa_{M}-\kappa_{D}\right)\right)+\left(1-\psi \right)\right)+\left(1-\chi \right)},$$
respectively. Transmission does not occur if the vector is preprandially killed or disarmed.

The rates of the preprandial and postprandial killing effects of tools on any given day *s* are calculated as
(7)$${\hat{\gamma }}_{pre}\left(s\right)=P_{\alpha }\left(s\right)\frac{\chi \psi \phi \kappa_{M}}{\chi \left(\psi \left(1-\phi \left(\pi -\kappa_{M}-\kappa_{D}\right)\right)+\left(1-\psi \right)\right)+\left(1-\chi \right)}\;\mathrm{and}$$
(8)$${\hat{\gamma }}_{post}\left(s\right)=P_{\alpha }\left(s\right)\frac{\chi \psi (1-\phi \pi )\xi }{\chi \left(\psi \left(1-\phi \left(\pi -\kappa_{M}-\kappa_{D}\right)\right)+\left(1-\psi \right)\right)+\left(1-\chi \right)},$$
respectively. The probability of these events are considered to be Markovian; therefore, the probabilities of a vector being killed preprandially or postprandially by a tool before a host-encountering event on day τ given it feeds on day *t* are
(9)$$P_{\kappa_{M}}\left(t,\tau \right)=1-\exp\left(\sum_{m=t+1}^{\tau }{\hat{\gamma }}_{pre}\left(m\right)\right)\;\mathrm{and}$$
(10)$$P_{\xi }\left(t,\tau \right)=1-\exp\left(\sum_{m=t}^{\tau -1}{\hat{\gamma }}_{post}\left(m\right)\right),$$
respectively. Here, the vector can be killed preprandially while encountering the host on day τ.

The vectorial capacity on day *t* is then calculated as the probability the vector feeds on a host on day *t* and survives until the end of the day, multiplied by the probability it infects another host every day after, given it survives until that day:(11)$$VC\left(t\right)=P_{h\to v}\left(t\right)P_{\mu }\left(t,t\right)\left(1-P_{\xi }\left(t,t\right)\right)\times \sum_{x=t+1}^{\infty }P_{v\to h}\left(t,x\right)P_{\mu }\left(t,x\right)\left(1-P_{\kappa_{M}}\left(t,x\right)\right)\left(1-P_{\xi }\left(t,x-1\right)\right).$$

### 2.2. Model Application: Arbovirus Transmission and Control in the UK

#### 2.2.1. Parameterising the Model for *Culicoides*-Borne Viruses

We parameterise the model for BTV, AHSV, EHDV, and SBV transmission in the UK facilitated by *C. obsoletus*, the same vector considered to transmit BTV in the Brand and Keeling [13] model. Here, the model will have a daily time step. Since we consider transmission by the same vector for all viruses, the parameters for the gonotrophic cycle completion rate and vector mortality rate are the same. The rate of gonotrophic cycle completion was parameterised by Mullens et al. [9] as
(12)$$\hat{\alpha }\left(s\right)=\max\left(0,1.9\times 10^{-4}T\left(s\right)\left(T\left(s\right)-3.7\right){\left(41.9-T\left(s\right)\right)}^{1/2.7}\right),$$
where *T* is the temperature in degrees Celsius on day *s*. For the rate of mortality, we use the reciprocal of the vector lifespan
(13)$$\hat{\mu }\left(s\right)=\frac{1}{\left(111.84\exp(-0.1547T(s))\right)},$$
parameterised by Gerry and Mullens [28].

Carpenter et al. [12] parameterised a model for the EIP completion rate for BTV, AHSV and EHDV, as well as the probability of transmission from host to vector. This model assumes that
(14)$$\hat{\sigma }\left(s\right)=\max\left(0,\alpha (T\left(s\right)-T_{min})\right),$$
where $T_{min}$ is the minimum threshold temperature that virus replication occurs and α is the rate of viral replication per degree-day above this threshold. The model was parameterised for five BTV studies. The model was only parameterised for *C. sonorensis* for AHSV and EHDV, and since there is no parameterisation for *C. obsoletus*, we use the mean of the three *C. sonorensis* experiments. These parameter values, along with parameters for the other virus, are given in Table 2.

Gubbins et al. [29] estimated α, $T_{min}$ and $\rho_{h\to v}$ for SBV using statistical inference for a within-farm model, utilising seroprevalence data for cattle and sheep farms in Belgium and The Netherlands. In this study, the probability of transmission from host to vector was $\rho_{h\to v}=0.14$. More recently, the susceptibility of field caught *Culicoides* in the UK was investigated by Barber et al. [30]. This study found that 12% of *Culicoides* contained significant quantities of SBV RNA within their heads 8 days post-feeding on viraemic blood through an artificial membrane. However, this value depends on the cycle threshold selected when analysing polymerase chain reaction data. For a more conservative cut-off level, this estimation decreased to 7%. We used $\rho_{h\to v}=0.12$, reflecting the less conservative value since it is closer to the Gubbins et al. [29] value, which was jointly estimated with other model parameters, and the values for other viruses in this study.

The parameters for the probability of transmission from vector to host ($p_{v\to h}$) and infectious period for each virus and host species are given in Table 3. There is limited data to inform this parameter for EHDV. Ruder et al. [31] experimentally infected two calves with EHDV by allowing infectious *C. sonorensis* to feed on them. As both animals became infectious, we assume that the probability of vector-to-host transmission is high, and assign the value estimated for BTV ($p_{h\to v}$ 0.9), as this is the highest value estimated amongst the other viruses in this study.

#### 2.2.2. Climate Data

The mean and maximum daily temperatures for the years 1973–2022 were extracted from the Hadley Centre Central England Temperature (HadCET) dataset [37], which are representative of a triangular region of the UK enclosed by Lancashire, London, and Bristol. The temperature-dependent rates were calculated for each day using the mean daily temperature.

#### 2.2.3. Comparing Vectorial Capacity of Viruses

First, we will consider scenarios with no vector-control tools present. In this case, the parameters π, $\kappa_{M}$, $\kappa_{D}$ and ξ are set to 0. In this scenario, the probability the vector feeds is
(15)$$P_{\alpha }\left(\tau \right)=1-\exp\left(-(\epsilon \hat{\alpha }\left(\tau \right))\right).$$

The equation for vectorial capacity also simplifies to
(16)$$VC\left(t\right)=P_{h\to v}\left(t\right)P_{\mu }\left(t,t\right)\times \sum_{x=t+1}^{\infty }P_{v\to h}\left(x\right)P_{\mu }\left(t,x\right),$$
which, if the blood index is set to 1, reproduces results from the Brand and Keeling [13] model.

Host selection is informed by a systematic literature search for the sources of blood meals in *Culicoides* [38]. Here, we use the average percentage of blood meals which were from the competent host, given the host was present at a location. For the *C. obsoletus* complex, the average for these values in Europe were 0.42, 0.56, and 0.41, for cattle, horse, and deer, respectively. This average was calculated as the mean weighted by the number of *Culicoides* caught per location.

The temperature data are divided into 10-year blocks: 1973–1982, 1983–1992, 1993–2002, 2003–2012 and 2013–2022. The vectorial capacity is calculated for each year. We then compare how the vectorial capacity has changed as the climate has changed according to the mean for each of the 10-year blocks.

For the 2013–2022 block, the mean number of days above the minimum threshold temperature that virus replication occurs ($T_{min}$) is calculated using the minimum and maximum daily temperatures for each virus.

#### 2.2.4. Uncertainty Analysis

The model simulated varying parameters individually to allow for the effects of perturbations in these parameters to be considered. Parameters considered varied in this analysis; their fixed values while varying other parameters and the ranges for which they are varied are given in Table 4. Here, the model is simulated for constant temperatures between 10 °C and 45 °C.

#### 2.2.5. Comparing $R_{0}$ of Viruses

Brand and Keeling [13] suggested a formula for $R_{0}$ based on the vectorial capacity. Here, for each day *t*, assuming a host becomes infectious on day *t*, $R_{0}$ is calculated as the sum of the vectorial capacity over the expected infectious period multiplied by the expected number of vectors per host. However, Brand and Keeling [13] used estimations for the expected number of vectors per host derived from field locations with no host present [39]. These values may not reflect the number of vectors feeding on each host per day.

Due to a lack of knowledge on the number of vectors per host, in this study, we do not attempt to calculate $R_{0}$. Instead, we calculate just the sum of the vectorial capacity over a host’s infectious period, without also multiplying by the expected number of vectors per host. The reciprocal of this value would predict the number of vectors per host required such that $R_{0}$ would be larger than 1. On each day *t*, we calculate the cumulative vectorial capacity from day *t* to day (*t* + host infectious period − 1). Here, it is important to note that we use vectorial capacity estimations that are scaled according to host selection; therefore, we are calculating the number of host-seeking vectors per competent host required, rather than the number of vectors required to bite each competent host.

Although the numbers of vectors in locations where each host type is found may vary; this method allows us to compare potential values of $R_{0}$ between viruses.

#### 2.2.6. Feeding Multiple Times per Gonotrophic Cycle

We simulate the model for the years 2013–2022, assuming that the vector feeds 1, 2 or 3 times per gonotrophic cycle. Here, we consider host selection in the model.

#### 2.2.7. Vector Control

Baker et al. [40] showed that untreated netting could reduce the number of *Culicoides* in the *C. obsoletus* group entering a horse stable by 75% (π=0.75). Mortality during World Health Organisation (WHO) cone bioassays across trialled net treatments varied substantially between days post-treatment and the treatments trialled. Effectiveness of interventions declined to less than 40% mortality after 7 days in most and 14 days for almost all interventions studied. We therefore simulate the model for 20% and 40% mortality induced by contact with the net. This is assumed to occur both on entrance and exit of the stable, and therefore is both a preprandial and postprandial killing effect.

We simulate the model for AHSV with 25%, 50% and 75% coverage (ψ=0.25, 0.50, 0.75) and 50% and 90% usage (ϕ=0.50, 0.90) for each proposed netting scenario in Table 5. As well as the varied mortality rates, we also simulate the model for an untreated net and a net which both kills and disarms. Previous studies have shown that as some insecticides age, their modes of action change, and they can disarm rather than kill [25].

## 3. Results

### 3.1. Model Development

Our study introduces a novel mathematical model for vectorial capacity that integrates both climatic factors and vector control interventions. This model builds upon existing frameworks by incorporating several key innovations. Firstly, it considers the impact of climate on various aspects of vector biology, including the gonotrophic cycle completion rate, vector mortality rate, and pathogen incubation rate. Secondly, it explicitly accounts for the effects of vector control measures at different stages of the vector feeding cycle, allowing for a more comprehensive assessment of intervention strategies. Thirdly, the model incorporates host availability and vector preferences, providing a more realistic representation of disease transmission dynamics. Finally, it allows for the consideration of vectors feeding multiple times per gonotrophic cycle, a behaviour observed in some vector species that can significantly impact disease transmission. This integrated approach enables a more comprehensive analysis of arbovirus transmission potential under varying climatic conditions and control scenarios, as demonstrated in the following results.

### 3.2. Comparing Vectorial Capacity of Viruses

Figure 1 shows the influence of the change in climate on the vectorial capacity. We observe that, for all diseases, the predicted vectorial capacity tends to be greater for more recent years. For EHDV, our calculations show that the vectorial capacity is only non-negligible for the most recent 2013–2022 block.

We observe that considering host selection reduces the vectorial capacity. Therefore, not considering this in models may lead to overestimating the ability of vectors to transmit a disease.

Overall, AHSV consistently has the largest vectorial capacity, compared to the other diseases, for both the non-scaled and scaled models. When comparing AHSV to BTV and SBV, this is due to the probability of transmission from host to vector being much more likely for AHSV. This probability is largest for EHDV; however, this virus has a much larger minimum threshold temperature for replication than the other viruses in this study. This is why there is a relatively short time-span where the vectorial capacity of EHDV is above 0 in the UK. However, here we have used the mean daily temperature for our calculations. There are likely to be days which are therefore assumed to have no virus replication because the mean temperature does not exceed the minimum threshold for virus replication, whereas in fact the maximum temperature does exceed this value. Table 6 shows that, for all viruses, there are more days with potential replication when the maximum daily temperature is considered, compared with the mean. This is particularly the case for EHDV, for which on average there were nearly seven times more days when the maximum temperature exceeded the replication threshold, compared to the mean temperature.

### 3.3. Uncertainty Analysis

Figure 2 shows the uncertainty analysis results. We observe that the model is not sensitive to small perturbations in the parameters describing the EIP completion rate ($T_{min}$ and α). Similarly, the model is not very sensitive to small changes in the probability of transmission from host to vector or vector to host ($\rho_{h\to v}$ or $\rho_{v\to h}$). Therefore, the model’s outputs are likely to remain robust, even with minor inaccuracies in these parameter estimations. Although the same hold for moderate uncertainty around the blood index of susceptible hosts, the model is much more sensitive to this parameter. Within the range of the means for hosts in this study (0.41–0.56), small inaccuracies will not significantly effect the model output. However, this highlights the importance of considering the presents of other non-susceptible hosts when considering the capacity for transmission of these viruses.

### 3.4. Comparing $R_{0}$ of Viruses

Although the predicted vectorial capacity is found to be largest for AHSV when compared with BTV and EHDV, it has a relatively short host infectious period (due to high mortality rates). Figure 3 shows that when we look at the sum of the vectorial capacity over the infectious period of a host, given it became infectious on day *t*, the cumulative vectorial capacity over this period is comparable for AHSV and BTV. Results suggest that EHDV could also have the same cumulative vectorial capacity as SBV for a short period during the summer months in the UK.

If we consider the reciprocal of the sum of the vectorial capacity over the infectious period of the host, we note that these predictions suggest that around three host seeking vectors per host type of interest would result in the peak $R_{0}$ being above 1 for BTV and AHSV, whereas for SBV and EHDV, the prediction would be around 20 host-seeking vectors per host.

### 3.5. Feeding Multiple Times per Gonotrophic Cycle

Figure 4 demonstrates that, if vectors are feeding on hosts more than one time per gonotrophic cycle, the vectorial capacity of these viruses could be much larger. This increase is non-linear, since there is an increased probability each day of the vector both acquiring the virus and transmitting the virus once it becomes infectious.

### 3.6. Vector Control

Figure 5 shows the predicted reduction in vectorial capacity due to the four types of stable netting described in Table 5. For each netting type, the reduction in vectorial capacity is greater with higher coverage and usage/adherence. Predictions suggest that untreated netting could reduce vectorial capacity by approximately 70% with high coverage (75%) and usage (90%); however, all treated netting options are competitive with this, even at low coverage (25%) and usage (50%). With coverage or usage below 50%, untreated netting does not substantially reduce vectorial capacity.

Netting that induces 40% mortality on contact reduces vectorial capacity by over 80% for all coverage and usage scenarios. Netting that disarms 20% of vectors and kills 20% on contact does not significantly reduce the vectorial capacity compared to interventions that only kill 20% of vectors on contact. This suggests that the mode of action transitioning to disarming from mortality as the intervention ages, or as vectors become resistant, would not support continual reductions in vectorial capacity.

For vectors that feed multiple times per feeding cycle, netting with modes of action beyond repelling have increased reductions in vectorial capacity. This is because the vector potentially comes into contact with the netting more times per feeding cycle, increasing the probability of mortality before it becomes, and once it is, infectious.

## 4. Discussion

We developed a model for vectorial capacity that considers both climate and host selection behaviour of vectors and vector controls. This model can be applied to more accurately predict vectorial capacity with current, or potential future, vector control strategies. Although temperature was the only climatic effect considered in this study, this model allows for additional climatic variables to be considered. This study focused on the vectorial capacity of *Culicoides*-borne viruses; however, it can be utilised for other pathogens and vectors.

We calculated the vectorial capacity based on a singular estimate for mean temperature. However, temperature varies according to location. Even within relatively small areas, microclimates can affect the temperature and vectorial capacity [21,41,42]. Previous studies indicate that landscape can significantly influence disease seroprevalence [43]. However, this variation in disease seroprevalence across different landscapes could also be influenced by the varying abundance of vectors. This, in turn, may depend on the thermal preferences of the vectors [44].

When considering the mean daily temperature, the number of days above the minimum threshold temperature for virus replication within the vector can be much lower than for the daily maximum temperature, especially for EHDV. Models using the mean temperature may therefore underestimate the vectorial capacity and the potential for pathogen transmission. Future work could explore more nuanced methods to estimate the duration of potential virus transmission, such as accumulated degree days above the threshold temperature, to incorporate within-day temperature variation. However, as finer-scale temperature data are not commonly reported, future studies could focus on developing and validating proxy measures or statistical methods to estimate accumulated degree days from available daily minimum, maximum, and mean temperatures. Such approaches could significantly improve our ability to predict virus transmission potential across varying climatic conditions.

We showed that considering host selection, driven by host availability and preferences of vectors, in models can significantly reduce vectorial capacity predictions. Therefore, the availability and preferences for competent and non-competent hosts, the availability of which varies by location, should be considered when assessing risk and deciding where to implement vector control.

In this study, $R_{0}$ was not calculated due to uncertainty of the vector-landscape. Two viruses may have the same vectorial capacity but different values of $R_{0}$ due to a different number of vectors per host, which will be variable across location and host type [43,45]. Brand and Keeling [13] used the number of *Culicoides* caught in light-traps without hosts present to parameterise the model. However, light-traps have been shown to not be good surrogate hosts for *Culicoides* [45,46].

The peak vectorial capacity summed over the infectious period for AHSV and EHDV were similar to BTV and SBV, respectively. Given that the UK has previously seen outbreaks of BTV and SBV, this study suggests that AHSV and EHDV outbreaks could also be a threat. Recently, in 2024, recommencement of direct horse movements from South Africa to the European Union has been approved [47]. Our results suggest that an importation of an infected case could lead to further transmission. This study provided evidence that vectorial capacity of the viruses the model was applied to in this study has been increasing in the UK. This highlights the importance of preparedness for the potential emergence of these diseases in the UK and the rest of Europe.

In dividing the vectorial capacity estimates into ten-year blocks and calculating averages for each period, we aimed to observe long-term trends in vectorial capacity. However, this methodological choice introduces the potential for discontinuities between the blocks. Future analyses might benefit from applying smoothing techniques. Despite this, the model predicted that the vectorial capacity of EHDV was only significantly above 0 for the most recent block, ending in 2022. The first reported case of EHDV in Europe was 2022, and outbreaks have continued to occur [48,49].

Brand and Keeling [13] compared three models of vectorial capacity: deterministic, Markovian, and intermediate. The intermediate model requires knowledge of the distribution of the EIP by setting a shape parameter for the gamma distribution (*k*). Here, we simulated the Markovian model (by assuming k=1 in Equation (2)), as *k* is not known for all viruses in the study. However, the model developed allows the EIP be modelled using the deterministic or intermediate model from Brand and Keeling [13]. If these other models are used, it is important to consider that the vectorial capacity predictions will generally be lowest for the deterministic model and largest for the Markovian model, with the intermediate model falling in between. Therefore, direct comparisons can not be made between results for different climates/viruses if a different model is used.

This study focused on adulticide vector-control tools. However, other tools, such as larvicide, have been shown to be effective against *Culicoides* [50]. Previously, the killing of larvae has been integrated into models of vectorial capacity by multiplying the vectorial capacity by the coverage of larvicide; however, larvicide does not affect vectorial capacity. By definition, vectorial capacity is the number of cases which arise from a single day of feeding on an infectious host, therefore vectors must have made it to the adult stage to influence vectorial capacity. Instead, larviciding impacts the number of vectors per host, hence reducing $R_{0}$. If the Brand and Keeling [13] method is used to calculate $R_{0}$ from the vectorial capacity, proportionally reducing vectorial capacity or the number of vectors per host by larvicide coverage would yield the same result.

Usage/adherence is sometimes not considered in vector-control studies; however, this study shows that they can be just as important as coverage for reducing vectorial capacity. This is expected, since if a tool is unused, it will not have an effect. A drawback is that this parameter can be hard to quantify. Usage describes a reduction in efficacy due to behaviours such as not stabling the horse every night or whether the horse is outside the stable for some of the time vectors are host seeking. However, vector feeding behaviours are not consistent between nights or over the duration of a night [51].

The data used to parameterise the stable netting control strategy was gathered using WHO cone bioassays and light-traps within stables. Light traps have been observed to not effectively estimate protective efficacy for spatial repellents [52]. Considering this, and the relatively small number of vectors caught in the stable (n = 25), the estimation for the reduction in biting may be inaccurate. WHO cone bioassays have also been shown to induce more mortality compared to when the interventions are evaluated in semi-field or experimental-hut conditions [53,54]. In this study, we estimate the reduction in vectorial capacity for treated netting that disarms mosquitoes. However, it is important to note that this disarming behaviour was not directly observed during the trials that were used to parameterise the model. Despite these drawbacks, we were able to provide a framework for evaluating the potential implication of transmission for tools with a range of properties. We found that whilst untreated stable netting could need to have high coverage and usage to substantially reduce the vectorial capacity of AHSV in the UK, insecticide treated netting could potentially decrease vectorial capacity, even at low coverage and usage levels.

Other adulticides have been been shown to be effective against *Culicoides* for use on host species of relevance to the viruses in this study [55]. Robin et al. [56] found that there was no reduction in the percentage of *Culicoides* caught in traps which were blood-fed next to a horse treated with topical deltamethrin, compared to an untreated horse. However, the difference in the total number or number of blood-fed *Culicoides* was not reported. Killing the vectors before they feed (preprandial killing) would likely increase the percentage of blood-fed vectors caught, while disarming them would decrease it. In contrast, using repellents might not change this percentage, as the insects are still able to escape after encountering the repellent.

This study highlighted the potential added efficacy of vector control tools against mosquitoes that feed multiple times per gonotrophic cycle, compared to those who just feed once. There is evidence that successive blood meals after an infectious blood meal can increase vector competence and shorten the EIP of vectors [57,58,59,60,61]. Therefore, this behaviour could increase the infectiousness of vectors, which was not considered in this study. If feeding on multiple hosts were to occur due to disruption when feeding, the size, and therefore viraemic dose, of the blood meal may be less, which may also reduce the probability the vector becomes infectious [60].

We have established a novel model for vectorial capacity that considers the influence of both climate and vector control at each stage of the vector feeding cycle. We demonstrated how this model can be used to compare the vectorial capacity of viruses. This model was used to assess the potential for outbreaks by comparing the total vectorial capacity during a host’s infectious period between viruses that were previously unobserved in a region and those that had been observed there before. Extensions to this work could include global sensitivity analysis to explore the influence of tools across a range of vector bionomics or temperatures, using methods similar to [62] to identify target product profiles.

Our research provides valuable insights into how the strategic use of insecticide-treated netting, even at low coverage levels, can substantially reduce the transmission potential of AHSV. This model offers a powerful tool for policymakers and health professionals, aiding in the formulation of more effective vector management strategies that could mitigate the impact of these diseases, especially in the context of changing global climates.

## Figures and Tables

**Figure 1 viruses-16-01221-f001:**
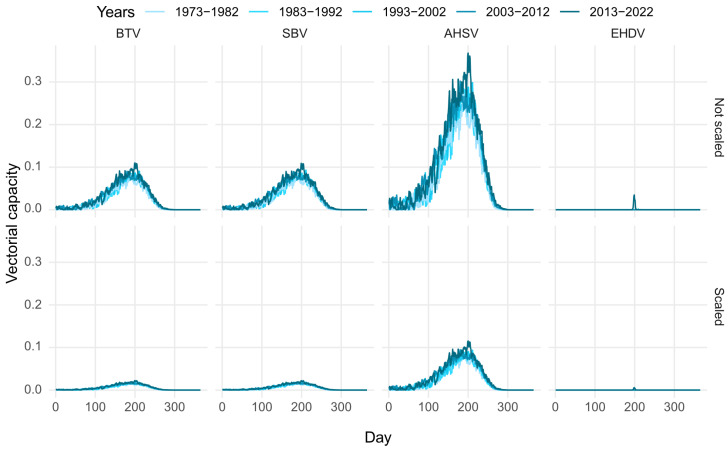
The predicted vectorial capacity for bluetongue virus (BTV) in cattle, Schmallenberg virus (SBV) in cattle, African horse sickness virus (AHSV) in horses, and epizootic haemorrhagic disease virus (EHDV) in deer; (**Top**) not scaled for host selection, (**Bottom**) scaled for host selection.

**Figure 2 viruses-16-01221-f002:**
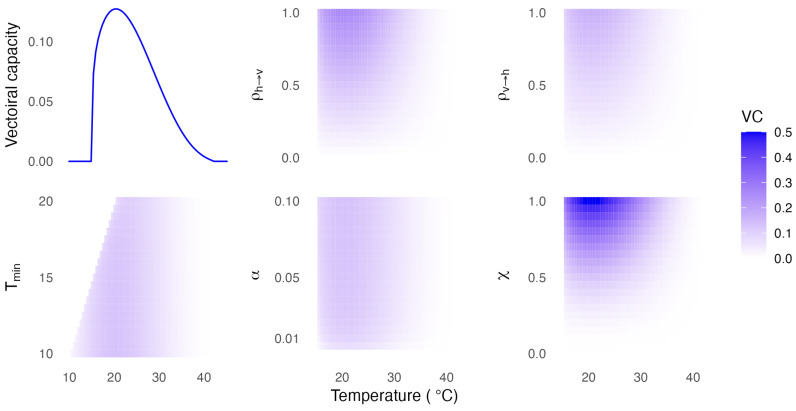
The vectorial capacity estimates using fixed parameter values and under constant temperatures (top left) and the uncertainty analysis outputs for each parameter analysed, as described in Table 4.

**Figure 3 viruses-16-01221-f003:**
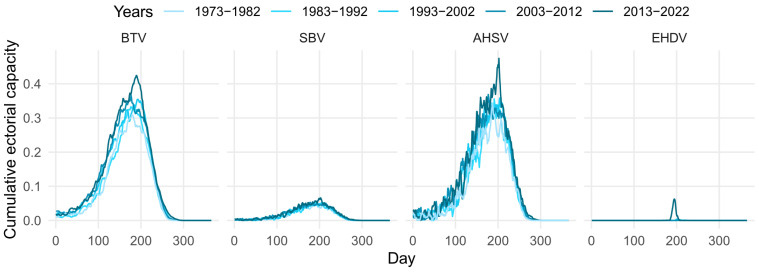
Mean vectorial capacity predictions summed over the duration of the infectious period for bluetongue virus (BTV) in cattle, Schmallenberg virus (SBV) in cattle, African horse sickness virus (AHSV) in horses, and epizootic haemorrhagic disease virus (EHDV) in deer.

**Figure 4 viruses-16-01221-f004:**
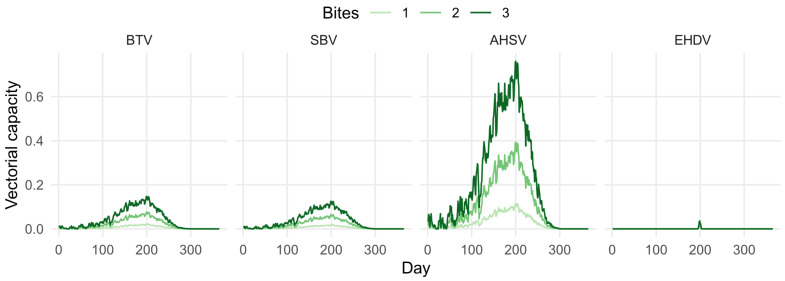
Mean vectorial capacity predictions for 2013–2022, considering host selection, for bluetongue virus (BTV), Schmallenberg virus (SBV), African horse sickness virus (AHSV), and epizootic haemorrhagic disease virus (EHDV), assuming vectors feed on average 1, 2, or 3 times per gonotrophic cycle.

**Figure 5 viruses-16-01221-f005:**
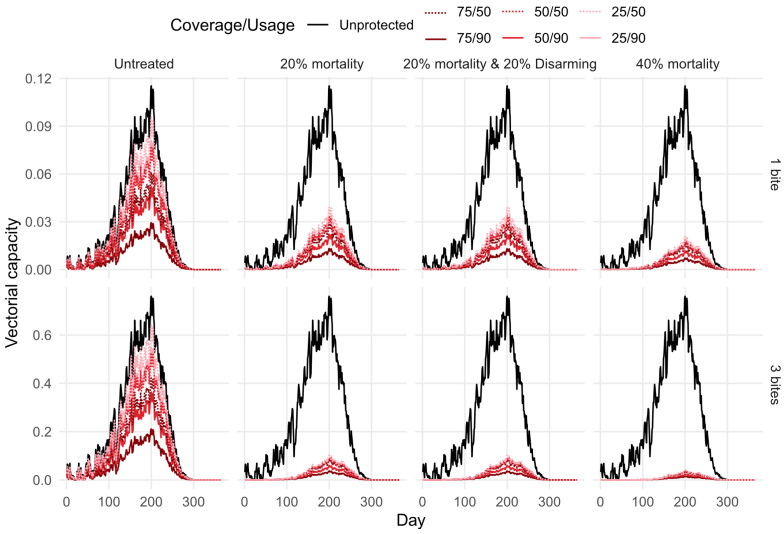
Mean vectorial capacity predictions for 2013–2022, considering host selection, for bluetongue virus (BTV), Schmallenberg virus (SBV), African horse sickness virus (AHSV), and epizootic haemorrhagic disease virus (EHDV), assuming various usage and coverage for four types of netting. Top: vectors are assumed to feed once per gonotrophic cycle. Bottom: vectors are assumed to feed an average of three times per gonotrophic cycle.

**Table 1 viruses-16-01221-t001:** Model parameter definitions. Parameters denoted with $\hat{\alpha }$ are vectors with elements of daily calculated variables dependent on climatic conditions. Above and below the filled line gives fundamental and composite parameters, respectively.

Parameter	Definition
$\hat{\alpha }\left(s\right)$	Rate of gonotrophic cycle completion on day *s*.
$\hat{\mu }\left(s\right)$	Vector mortality rate on day *s*.
$\hat{\sigma }\left(s\right)$	Rate of pathogen EIP completion on day *s*.
χ	Proportion of blood meals from competent hosts, referred to as the blood index.
ϵ	Average number of bites per vector per gonotrophic cycle.
π	Reduction in the rate of vector biting due to the presence of a vector-control tool.
$\kappa_{M}$	Increase in the rate of vector mortality before biting due to the presence of a vector-control tool, relative to the rate of biting without the vector-control tool.
$\kappa_{D}$	Rate of vector disarming due to the presence of a vector-control tool, relative to the rate of biting without the vector-control tool.
ξ	Increased probability of vector mortality after biting due to the presence of a vector control tool.
ψ	Proportion of the target hosts with access to the vector-control tool, referred to as coverage.
ϕ	Adherence to using the vector-control tool, referred to as usage.
$\rho_{h\to v}$	Probability of transmission from host to vector, given that the host is infectious.
$\rho_{v\to h}$	Probability of transmission from vector to host, given that the vector is infectious.
$P_{\mu }\left(t,\tau \right)$	Probability a vector which fed on day *t* survives until day τ.
$P_{\sigma }\left(t,\tau \right)$	Probability that a vector which fed on day *t* is infectious day τ, given transmission occurred during the bite on day *t*.
$P_{\alpha }\left(s\right)$	Probability of a host-encountering event on day *s* (defined as feeding or preprandial mortality or disarming due to the presence of a tool).
$P_{h\to v}\left(t\right)$	Probability of transmission of a pathogen from host to vector on day *t*.
$P_{v\to h}\left(\tau \right)$	Probability of transmission of a pathogen from vector to host on day τ.
${\hat{\gamma }}_{pre}\left(s\right)$	Rate of preprandial killing on day *s*.
${\hat{\gamma }}_{post}\left(s\right)$	Rate of postprandial killing on day *s*.
$P_{\kappa_{M}}\left(t,\tau \right)$	Probability a vector is killed preprandially by day τ given it being fed on day *t*.
$P_{\xi }\left(t,\tau \right)$	Probability a vector is killed postprandially by day τ given it being fed on day *t*.
VC(t)	The vectorial capacity on day *t*.

**Table 2 viruses-16-01221-t002:** The parameters for the minimum threshold temperature that virus replication occurs ($T_{min}$), the rate of viral replication per degree-day above this threshold (α), and the probability of transmission from host to vector ($\rho_{h\to v}=0.14$) for bluetongue virus (BTV), African horse sickness virus (AHSV), Schmallenberg virus (SBV), and epizootic haemorrhagic disease virus (EHDV).

Virus	$T_{\min}$	α	$\rho_{h\to v}$	Ref.
BTV	12.6	0.019	0.13	[12]
AHSV	12.6	0.017	0.52	[12]
SBV	12.35	0.03	0.12	[29,30]
EHDV	19.5	0.084	0.92	[12]

**Table 3 viruses-16-01221-t003:** The parameters for the transmission probability from vector to host ($\rho_{h\to v}$) and host infectious period for bluetongue virus (BTV), African horse sickness virus (AHSV), Schmallenberg virus (SBV), epizootic haemorrhagic disease virus (EHDV), and their host species. †: Assumed to be the same as BTV.

Virus	Host Species	$\rho_{v\to h}$	Infectious Period	Ref.
BTV	Cattle	0.9	20.6	[32,33]
AHSV	Horse	0.77	4.4	[34]
SBV	Cattle	0.76	3.04	[29]
EHDV	Deer	0.9 †	11.7	[35,36]

**Table 4 viruses-16-01221-t004:** Parameters considered in the uncertainty analysis, their fixed value while varying other parameters, and the ranges for which they are varied.

Parameter	Fixed Value	Range
$\rho_{h\to v}$	0.50	0–1
$\rho_{v\to h}$	0.75	0–1
$T_{min}$	15	10–20
α	0.05	0.01–0.10
χ	0.50	0–1

**Table 5 viruses-16-01221-t005:** Modes of action of stable netting scenarios simulated. Full descriptions of parameters are given in Table 1.

Scenario	Bite Rate Reduction (π)	Relative Preprandial Mortality Rate Increase ($\kappa_{M}$)	Relative Disarming Rate Increase ($\kappa_{D}$)	Postprandial Mortality Increase (ξ)
Untreated	0.75	0.00	0.00	0.00
20% mortality	0.75	0.20	0.00	0.20
40% mortality	0.75	0.40	0.00	0.40
20% mortality & 20% disarming	0.75	0.20	0.20	0.20

**Table 6 viruses-16-01221-t006:** The mean number of days per year for 2013–2022 with temperature above the minimum temperature threshold for virus replication for bluetongue virus (BTV), Schmallenberg virus (SBV), African horse sickness virus (AHSV), and epizootic haemorrhagic disease virus (EHDV), and the mean and maximum daily temperatures across Central England.

Virus	Threshold Temperature	Days above Threshold Temperature	
		Mean Temperature	Maximum Temperature
BTV	12.6	136.4	213.7
SBV	12.4	140.4	217.5
AHSV	12.6	136.4	213.7
EHDV	19.5	11.4	79.4

## Data Availability

R code to run the model is available at https://github.com/emmafairbanks/EntoModels (DOI:10.5281/zenodo.12785165).

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
