# Peer review of "Modelling the Influence of Climate and Vector Control Interventions on Arbovirus Transmission"

_viruses, 2024, doi:10.3390/v16081221_

Round 1

Reviewer 1 Report

Comments and Suggestions for Authors

Dear Authors,

Thank you very much for the opportunity to review your manuscript. I found your model to be very interesting, and your manuscript quite clear. However, I do have have some questions and suggestions that might benefit your paper:

1. Reading your Results section, I kept wondering if the model predictions, especially for the 2013-2022 block, were matching up with actual experience. Later on, in the Discussion, I learned that the UK has experienced outbreaks of BTV and SBV, but not AHSV or EHDV. Can you please talk about previous outbreaks in the study area in your Introduction. It would help the reader not to wonder until the Discussion.

2. At the bottom of page 3, in your Results, you talk about Table 1 showing how your results in terms of the number of days with/without viral replication are quite sensitive to the choice of either using the minimum, mean or maximum daily temperature to compare with the temperature threshold for replication. I was wondering if using something like an accumulated degree-minutes would be meaningful (like the number of minutes in a day that exceed the viral replication threshold), to incorporate within-day temperature variation.

3. The caption for Table 1 has a duplicated "the" in the last line just after "(EHDV)".

4. Based on your results, in particular Figure 2 on the cumulative vectorial capacity, and the number of host seeking vectors required for R0 to be higher than 1, AHSV and BTV seem to be more likely to have outbreaks, compared to SBV and EHDV. However, as you mentioned, there have been outbreaks of BTV and SBV in the UK, but not AHSV or EHDV, even tough SBV has a lower likelihood than AHSV. How come there haven't been outbreaks of AHSV given these results? Can you comment on that in the Discussion?

 5. I was slightly confused when looking at Figure 3, because it came after Figure 2. Figure 2 shows cumulative vectorial capacity (~R0), and so when I looked at Figure 3, I was expecting the same. However, Figure 3 shows just vectorial capacity, not cumulative. The only thing clarifying that (other than careful reading of the caption, is the y axis label. Would it be possible to emphasize that this Figure is showing just vectorial capacity, not cumulative? One way to do that might be to switch Figure 2 and 3, but then you'd have to switch the order of the text as well. Or show cumulative vectorial capacity on both Figure 3 and 4 instead of just simple vectorial capacity. Wouldn't that be more useful anyway?

6. On page 8, in the discussion, in the 4th paragraph, you talk about host selection. Do you mean selecting competent-vs-non-competent hosts? Perhaps putting "competent host selection" instead of just "host selection" would make that more clear.

7. In the last paragraph of page 8, you talk about using average temperatures in the 10-year blocks, and future analyses using smoothing techniques. How about just using the actual temperature data at an annual resolution? That would incorporate a lot more variability, but also make it more explicit.

8. In your methods, there are a number of terms that I was unfamiliar with. One of these were "prepandrial" and "postpandrial", which I came to understand as "before feeding" and "after feeding". Can you please define those when you first use them?

9. It took me a while to understand that you model is not the traditional compartmental type model that I'm more familiar with, and that you do not keep track of the number of susceptible/infectious vectors and susceptible/infectious/recovered hosts of any kind. If I understand correctly, all you're doing is running functions that ultimately calculate vectorial capacity, which is defined as the number of susceptible hosts that an infectious vector will infect until it dies. I'm still a little bit confused on how you can calculate this without knowing how many infectious and susceptible hosts you have in the population. Are you assuming that there are unlimited numbers of susceptible and infectious hosts? If yes, how realistic is that? Can you clarify this for me? Perhaps a paragraph at the beginning of the Methods to clarify your modeling approach would be helpful. 

10. On page 12, you mention that "most bites occur at the end of the day". Does that mean that Culicoides midges only bite at dusk, not at dawn? Can you provide a reference to that?

11. Another term that I was unfamiliar with is "disarming". What does it mean when a vector is disarmed? How is different from being killed? In your equations (4-8), it seems like killing and disarming by the vector control tool is doing the same thing. Are they the same? Then why keep them separate? I see killing involved in the pre-pandrial mortality in Eq. (7), but not disarming. Is that the only difference?

12. I also wonder about the functional importance of competent vs non-competent hosts. In Eq. (4), only the competent hosts seem to have protection measures applied to them. Is that an assumption of the model? If yes, please state that.

13. On page 12, right before Eq. (4), did you mean "host-encountering event" when you wrote "host-countering an event"?

14. In Eqs. (5-8), why do you use "s" for the day instead of "t" or "tau"? Does that just indicate any day between t and tau?

15. In Eqs. (5-6), the ratio seems to calculate the probability of biting a competent host over a non-competent host. However, the term in the numerator for the competent host does not feature the killing and disarming effect, while the one in the denominator does. Why is that? Is that because it doesn't matter if the vector survives for this ratio? But why?

16. Does Eq. (5) assume that the host is infectious, and Eq. (6) assume that the vector is infectious?

17. Does your model assume that only a single bite can occur per day? I understand that multiple bites can occur per gonotrophic cycle, but can there be multiple bites per day? Do I understand correctly that your model has a daily timestep?

18. Does your model assume that the probability of survival for midges per day is constant, and independent of the age of the midge? Is there evidence for that? 

19. It's a little confusing that your probability of survival is denoted by "mu". It suggests to me that it is the probability of mortality, and then I get confused that it is not featured as (1-P_mu) in the equations, as the probability of survival, like the other probabilities of survival and mortality are.

20. Throughout the paper, you use temperature values that are in Celsius, which is great. However, could you mention at some point, perhaps after Eq. (12) that T is in Celsius, just for clarity?

21. On page 14, in the first paragraph, after Eq. (14), please put "since" before "there is no parameterisation".

22. At the end of the same paragraph, do all these parameters come from Carpenter et al. [12]? If yes, can you repeat that reference for clarity?

23. In the next paragraph, please capitalize "culicoides".

24. In the same paragraph, fix "invecstigated".

25. I really like Table 3 with the relevant parameters for each virus. Can you include the other parameters you list on page 14 in this Table as well, with their citations, such as alpha, Tmin and the host-to-vector probability of transmission? Then you could just refer to Table 3 from the text. 

26. On page 15, in the paragraph after Eq. (16), you talk about host selection. Can you again add "competent" to host selection, if that's what you mean?

27. On page 16, second para, you describe how the reciprocal of the sum of the vectorial capacity over a host's infectious period would predict the number of vectors per host that would be required such that R0 would be larger than 1. Why is that? Can you explain in more detail? I kind of get it, but not really. So does that mean that you actually calculate R0? If not, why not?

28. There are a number of entries in your References where species names are not italicized ([7] and [53]), and capitalize "states" in [54].

Thank you very much again for allowing me to read your manuscript, and I hope you find my comments and questions useful! 

Author Response

  1. Reading your Results section, I kept wondering if the model predictions, especially for the 2013-2022 block, were matching up with actual experience. Later on, in the Discussion, I learned that the UK has experienced outbreaks of BTV and SBV, but not AHSV or EHDV. Can you please talk about previous outbreaks in the study area in your Introduction. It would help the reader not to wonder until the Discussion.
    We have added “This model is then applied to temperature data from the UK, which has previously seen BTV and SBV outbreaks.” To the introduction. We have also added details on outbreaks of EDHV in Europe matching results.
  2. At the bottom of page 3, in your Results, you talk about Table 1 showing how your results in terms of the number of days with/without viral replication are quite sensitive to the choice of either using the minimum, mean or maximum daily temperature to compare with the temperature threshold for replication. I was wondering if using something like an accumulated degree-minutes would be meaningful (like the number of minutes in a day that exceed the viral replication threshold), to incorporate within-day temperature variation.
    The authors agree that this would be a great method and with sufficient data we would be keen to implement such a method. We have added some text on this to the discussion:
    “Future work could explore more nuanced methods to estimate the duration of potential virus transmission, such as accumulated degree-days above the threshold temperature, to incorporate within-day temperature variation. However, as finer-scale temperature data are not commonly reported, future studies could focus on developing and validating proxy measures or statistical methods to estimate accumulated degree-days from available daily minimum, maximum and mean temperatures. Such approaches could significantly improve our ability to predict virus transmission potential across varying climatic conditions.”

  3. The caption for Table 1 has a duplicated "the" in the last line just after "(EHDV)".
    Done.

  4. Based on your results, in particular Figure 2 on the cumulative vectorial capacity, and the number of host seeking vectors required for R0 to be higher than 1, AHSV and BTV seem to be more likely to have outbreaks, compared to SBV and EHDV. However, as you mentioned, there have been outbreaks of BTV and SBV in the UK, but not AHSV or EHDV, even tough SBV has a lower likelihood than AHSV. How come there haven't been outbreaks of AHSV given these results? Can you comment on that in the Discussion?
    This is essentially because SBV emerged in Europe, whereas AHSV is mostly restricted to endemic regions in Africa with strict movement restrictions from this country. We appreciate that most of the introduction and discussion is based around the model developed, rather than the application. We have added some further context for the application. These include reference to the recent approval of direct horse movements from South Africa to the EU and the first cases of EHDV in Europe in 2022. See paragraph 6 and 7 in the discussion.

  5. I was slightly confused when looking at Figure 3, because it came after Figure 2. Figure 2 shows cumulative vectorial capacity (~R0), and so when I looked at Figure 3, I was expecting the same. However, Figure 3 shows just vectorial capacity, not cumulative. The only thing clarifying that (other than careful reading of the caption, is the y axis label. Would it be possible to emphasize that this Figure is showing just vectorial capacity, not cumulative? One way to do that might be to switch Figure 2 and 3, but then you'd have to switch the order of the text as well. Or show cumulative vectorial capacity on both Figure 3 and 4 instead of just simple vectorial capacity. Wouldn't that be more useful anyway?
    In this research we focus on vectorial capacity because it is a well-defined, commonly considered value within the field of vector-borne diseases. See the response to comment 27 for clarifying the difference between the cumulative vectorial capacity and R0. Cumulative vectorial capacity is not common and is therefore not easily comparable to other models for vectorial capacity. Here, it is used to demonstrate how the length of the infectious period of the host also influences disease transmission and to be able to interpret possible values of R0 if the vectors per host was known. We prefer to continue to use vectorial capacity in all figures apart from the one displaying cumulative vectorial capacity. It is stated in the text, axis label and caption that here we are looking at the standard vectorial capacity.

  6. On page 8, in the discussion, in the 4th paragraph, you talk about host selection. Do you mean selecting competent-vs-non-competent hosts? Perhaps putting "competent host selection" instead of just "host selection" would make that more clear.
    We have edited the sentence after this to: “Therefore availability and preferences for competent and non-competent hosts, availability of which varies by locations, should be considered when assessing risk and deciding where to implement vector control.”

  7. In the last paragraph of page 8, you talk about using average temperatures in the 10-year blocks, and future analyses using smoothing techniques. How about just using the actual temperature data at an annual resolution? That would incorporate a lot more variability, but also make it more explicit.
    This was what was done. The methods section states: “The temperature data are divided into 10-year blocks: 1973–1982, 1983–1992, 1993–2002, 2003–2012 and 2013–2022. The vectorial capacity is calculated for each year. We then compare how the vectorial capacity has changed as the climate has changed according to the mean for each of the 10-year blocks.”

    We have edited this sentence in the discussion to make this clearer. It now reads: In dividing the vectorial capacity estimates into ten-year blocks and calculating averages for each period, we aimed to observe long-term trends in vectorial capacity.

  8. In your methods, there are a number of terms that I was unfamiliar with. One of these were "prepandrial" and "postpandrial", which I came to understand as "before feeding" and "after feeding". Can you please define those when you first use them?
    Added. The text now reads: In the model, when vectors encounter a host they either bite, are killed preprandially (before feeding), are disarmed or seek another host. If they bite they may be killed postprandially (after feeding).

  9. It took me a while to understand that you model is not the traditional compartmental type model that I'm more familiar with, and that you do not keep track of the number of susceptible/infectious vectors and susceptible/infectious/recovered hosts of any kind. If I understand correctly, all you're doing is running functions that ultimately calculate vectorial capacity, which is defined as the number of susceptible hosts that an infectious vector will infect until it dies. I'm still a little bit confused on how you can calculate this without knowing how many infectious and susceptible hosts you have in the population. Are you assuming that there are unlimited numbers of susceptible and infectious hosts? If yes, how realistic is that? Can you clarify this for me? Perhaps a paragraph at the beginning of the Methods to clarify your modeling approach would be helpful. 
    This is a model for vectorial capacity not a dynamical model for disease transmission within a population. The number of infectious bites a vector can deliver is more of a confinement than the number of hosts. This is not considered in vectorial capacity models. As you can see the vectorial capacity never usually rises above one. These are two very different types of model, calculating different things.

  10. On page 12, you mention that "most bites occur at the end of the day". Does that mean that Culicoides midges only bite at dusk, not at dawn? Can you provide a reference to that?
    We have amended this section to: This is because most bites occur sunset and twilight (Meiswinkel and Elbers, 2016), therefore we assume the EIP only depends on climatic conditions during days after the bite has occurred. It's important to note that bites occurring in the early hours of the morning are typically counted as part of the previous day's activity. The temperature during the day affects whether Culicoides will be active and bite that night.

  11. Another term that I was unfamiliar with is "disarming". What does it mean when a vector is disarmed? How is different from being killed? In your equations (4-8), it seems like killing and disarming by the vector control tool is doing the same thing. Are they the same? Then why keep them separate? I see killing involved in the pre-pandrial mortality in Eq. (7), but not disarming. Is that the only difference?
    This term is defined when it is first used: This model considers both personal protection through reduction in feeding, as well as community protection through vector mortality and disarming (feeding inhibition due to intoxication). Both preprandial killing and disarming are included in the probability of leaving host seeking, but only preprandial killing leads to mortality.

  12. I also wonder about the functional importance of competent vs non-competent hosts. In Eq. (4), only the competent hosts seem to have protection measures applied to them. Is that an assumption of the model? If yes, please state that.
    We have added: In this study, we assume that vector control tools are applied only to competent hosts.

  13. On page 12, right before Eq. (4), did you mean "host-encountering event" when you wrote "host-countering an event"?
    Fixed.

  14. In Eqs. (5-8), why do you use "s" for the day instead of "t" or "tau"? Does that just indicate any day between t and tau?
    s is used for daily calculations which are often later summed from day t to tau, or t to tau – 1 depending on the variable, as detailed in the text. This is essentially assigned to variables which are not constricted to a set day, like t or tau. We have updated the text throughout to ensure consistency. This included changing the probability of transmission for host to vector and from vector to host beginning changed from s to t and tau, respectively.

  15. In Eqs. (5-6), the ratio seems to calculate the probability of biting a competent host over a non-competent host. However, the term in the numerator for the competent host does not feature the killing and disarming effect, while the one in the denominator does. Why is that? Is that because it doesn't matter if the vector survives for this ratio? But why?
    This is because there will not be transmission if the vector is killed before biting or disarmed. We have added the following sentence after the equation: “Transmission does not occur if the vector is preprandially killed or disarmed.”

  16. Does Eq. (5) assume that the host is infectious, and Eq. (6) assume that the vector is infectious?
    We have updated the definition of vectorial capacity in the first paragraph to make it clear that the host is infectious. Whether the vector is infectious depends on P_sigma(t,tau), the probability the vector is infectious on day tau.

  17. Does your model assume that only a single bite can occur per day? I understand that multiple bites can occur per gonotrophic cycle, but can there be multiple bites per day? Do I understand correctly that your model has a daily timestep?
    The model itself could be adapted to different time steps. For clarity we have added a sentence to the application section: Here, the model will have a daily time step.

  18. Does your model assume that the probability of survival for midges per day is constant, and independent of the age of the midge? Is there evidence for that?
    Yes, it does. This is the same as the Brand and Keeling model. Evidence on exactly how age effects mortality is limited.
  19. It's a little confusing that your probability of survival is denoted by "mu". It suggests to me that it is the probability of mortality, and then I get confused that it is not featured as (1-P_mu) in the equations, as the probability of survival, like the other probabilities of survival and mortality are.
    This was kept the same as the Brand and Keeling model denotations.

  20. Throughout the paper, you use temperature values that are in Celsius, which is great. However, could you mention at some point, perhaps after Eq. (12) that T is in Celsius, just for clarity?
    Added to the definition of T.

  21. On page 14, in the first paragraph, after Eq. (14), please put "since" before "there is no parameterisation".
    Added.

  22. At the end of the same paragraph, do all these parameters come from Carpenter et al. [12]? If yes, can you repeat that reference for clarity?
    Added.

  23. In the next paragraph, please capitalize "culicoides".
    Fixed.

  24. In the same paragraph, fix "invecstigated".
    Fixed.

  25. I really like Table 3 with the relevant parameters for each virus. Can you include the other parameters you list on page 14 in this Table as well, with their citations, such as alpha, Tmin and the host-to-vector probability of transmission? Then you could just refer to Table 3 from the text. 
    These parameters cannot be added easily to Table 3 (now Table 2), because Table 3 describes host dynamics and the headings are as such. There is also quite a lot of discussion around the parameterising of the model in section 2.2.1.

  26. On page 15, in the paragraph after Eq. (16), you talk about host selection. Can you again add "competent" to host selection, if that's what you mean?
    We have added competent host to the sentence below this: Host selection is informed by a systematic literature search for the sources of blood meals in Culicoides (Fairbanks, In preparation). Here, we use the average percentage of blood meals which were from the competent host, given the host was present at a location.

  27. On page 16, second para, you describe how the reciprocal of the sum of the vectorial capacity over a host's infectious period would predict the number of vectors per host that would be required such that R0 would be larger than 1. Why is that? Can you explain in more detail? I kind of get it, but not really. So does that mean that you actually calculate R0? If not, why not?
    This paragraph and the paragraph before have been reformatted for clarity. As stated in sentence before this paragraph, we do not have accurate estimations for the number of vector per host, which is required to predict R0. Therefore, we do not multiply by this to get R0.

  28. There are a number of entries in your References where species names are not italicized ([7] and [53]), and capitalize "states" in [54].
    Fixed.

Reviewer 2 Report

Comments and Suggestions for Authors

Authors presented an updated mathematical model that predicts the transmission of arboviruses considering the influence of both climate change and vector control strategies. They applied the model to analyze the dynamics of vectorial capacity of several arboviruses in Europe during the last 50 years. Using parameter values extracted from the literature, the authors demonstrated some quantitative features of these dynamics associated with the climatic factors, which could be important in epidemiological terms.

The manuscript is well written, and all necessary details of the methodology used in the study are clearly presented. All graphs are sufficiently informative, and conclusions are well supported by the results. I believe this work presents a good framework for the further development of more precise models in the field.

I have the following questions/comments:

1.     One of the results presented by the authors is a new model, so it should be introduced in the beginning of the Results section. As the mathematical details are present in Methods, the model can be shortly described in Results in words, with the emphasis on specific novelties.

2.     All calculations were done using the parameter values extracted from heterogeneous literature sources, so it would be interesting to see whether the conclusions are robust to possible perturbations of these values. Some sort of the parameter sensitivity analysis is desirable.

3.     Figures: Is it possible to add the solution of the basic model, which does not account for the temperature change, as a reference value?

4.     The authors simulate the climate change until 2022. It would be interesting to see what the model predicts for the future temperature dynamics taken, for example, from the various climate change scenarios presented in the IPCC reports.

Author Response

  1. One of the results presented by the authors is a new model, so it should be introduced in the beginning of the Results section. As the mathematical details are present in Methods, the model can be shortly described in Results in words, with the emphasis on specific novelties.
    We have added a subsection entitled Model development at the beginning of the results section. This reads:
    Our study introduces a novel mathematical model for vectorial capacity that integrates both climatic factors and vector control interventions. This model builds upon existing frameworks by incorporating several key innovations. Firstly, it considers the impact of climate on various aspects of vector biology, including the gonotrophic cycle completion rate, vector mortality rate, and pathogen incubation rate. Secondly, it explicitly accounts for the effects of vector control measures at different stages of the vector feeding cycle, allowing for a more comprehensive assessment of intervention strategies. Thirdly, the model incorporates host availability and vector preferences, providing a more realistic representation of disease transmission dynamics. Finally, it allows for the consideration of vectors feeding multiple times per gonotrophic cycle, a behavior observed in some vector species that can significantly impact disease transmission. This integrated approach enables a more comprehensive analysis of arbovirus transmission potential under varying climatic conditions and control scenarios, as demonstrated in the following results.

  2. All calculations were done using the parameter values extracted from heterogeneous literature sources, so it would be interesting to see whether the conclusions are robust to possible perturbations of these values. Some sort of the parameter sensitivity analysis is desirable.
    We have performed some uncertainty analysis for the parameters specific to viruses or hosts. This includes an additional table, figure and text in the new sections entitled sensitivity analysis.

  3. Figures: Is it possible to add the solution of the basic model, which does not account for the temperature change, as a reference value?
    This is done for a range of temperatures for each parameter influenced my temperature and vectorial capacity in the Brand and Keeling article. We additionally show the vectorial capacity for different temperatures for the set parameters in the uncertainty analysis discussed in response to comment 2. Note that constant temperatures will have constant results and therefore these are reported as a single constant value per temperature, rather than a constant output for a timeseries.

  1. The authors simulate the climate change until 2022. It would be interesting to see what the model predicts for the future temperature dynamics taken, for example, from the various climate change scenarios presented in the IPCC reports.
    This is out of scope for this study.

Round 2

Reviewer 2 Report

Comments and Suggestions for Authors

I appreciate the efforts made by authors to improve the manuscript.